# Modulation of Radiation Doses and Chimeric Antigen Receptor T Cells: A Promising New Weapon in Solid Tumors—A Narrative Review

**DOI:** 10.3390/jpm13081261

**Published:** 2023-08-14

**Authors:** Antonio Pontoriero, Paola Critelli, Federico Chillari, Giacomo Ferrantelli, Miriam Sciacca, Anna Brogna, Silvana Parisi, Stefano Pergolizzi

**Affiliations:** 1Radiation Oncology Unit, Department of Biomedical, Dental Science and Morphological and Functional Images, University of Messina, 98125 Messina, Italy; apontoriero@unime.it (A.P.); chillari.federico@gmail.com (F.C.); giacomo.ferrantelli@outlook.com (G.F.); miriamsciacca06@gmail.com (M.S.); silvana.parisi@unime.it (S.P.); stpergolizzi@unime.it (S.P.); 2Radiotherapy Unit, Medical Physics Unit, A.O.U. “G. Martino”, 98125 Messina, Italy; anna.brogna@polime.it

**Keywords:** CART, tumor microenvironment, immunotherapy, radiotherapy, solid tumor

## Abstract

Tumor behavior is determined by its interaction with the tumor microenvironment (TME). Chimeric antigen receptor (CART) cell therapy represents a new form of cellular immunotherapy (IT). Immune cells present a different sensitivity to radiation therapy (RT). RT can affect tumor cells both modifying the TME and inducing DNA damage, with different effects depending on the low and high doses delivered, and can favor the expression of CART cells. CART cells are patients’ T cells genetically engineered to recognize surface structure and to eradicate cancer cells. High-dose radiation therapy (HDRT, >10–20 Gy/fractions) converts immunologically “cold” tumors into “hot” ones by inducing necrosis and massive inflammation and death. LDRT (low-dose radiation therapy, >5–10 Gy/fractions) increases the expansion of CART cells and leads to non-immunogenetic death. An innovative approach, defined as the LATTICE technique, combines a high dose in higher FDG- uptake areas and a low dose to the tumor periphery. The association of RT and immune checkpoint inhibitors increases tumor immunogenicity and immune response both in irradiated and non-irradiated sites. The aim of this narrative review is to clarify the knowledge, to date, on CART cell therapy and its possible association with radiation therapy in solid tumors.

## 1. Introduction

Cancer is characterized by a high mutation rate that leads to a process of transformation from healthy cells to cancer cells [1].

The immune system can recognize malignantly transformed cells through a group of molecules found on the surface, called tumor-associated antigens (TAAs), that differentiate a tumor cell from a normal one [2]. The first TAA was identified in the 1990s in testis cancer (MAGEA1), reporting high tumor antigenicity and a promising immunotherapeutic target [3,4,5].

Immunotherapy is based on the ability of T cells to activate and bind cancer cells, enhancing the immune system to recognize cancer as a foreign invader. 

A new therapeutic approach which aims to prevent treatment resistance, reducing the tumor burden, is represented by adoptive cellular therapy (ACT). ACT, or cellular immunotherapy, is a form of immunotherapy based on a patient’s T cells, which are genetically engineered to recognize cancer cells and then reinfused back into the patient [6].

To date, there are three major modalities of ACT: tumor-infiltrating lymphocytes (TILs), genetically engineered T cell receptors (TCRs), and chimeric antigen receptor (CAR) T cells. 

TILs are a polyclonal population of lymphocytes that reside in the tumor with broad antigen recognition in tumor cells [7].

TCRs are engineered T cells that recognize HLA-presented peptides derived from the proteins of all cellular compartments [8].

CART cells represent one of the most promising approaches in tumor immunotherapy, especially in treating hematological tumors [9,10,11,12,13,14,15,16]. 

In solid tumors, efficacies are not demonstrated yet. The first therapeutic approach in solid tumors was reported in 2006 by Rosemberg. He found that TCR-T therapy could recognize the melanocyte-differentiating antigen (MART-1) with highly positive results in the treatment [17,18]; afterward, Cartellieri et al. showed that in melanoma, CART can significantly improve survival conditions without interruption [19].

The aim of this narrative review is to clarify the knowledge, to date, on CART cell therapy and its possible association with radiation therapy in solid tumors. 

## 2. CART: Chimeric Antigen Receptor (CAR) T Cells

In 2017, the FDA approved the first anti-CD19 CART cell product, tisagenlecleucel, for relapsed and/or refractory B-cell precursor acute lymphoblastic leukemia (B-ALL) in pediatric and young adult patients. 

CART involves lymphocytes entirely engineered to recognize any cell surface structure, independently of the MHC (major histocompatibility complex) presentation, leading to stronger T cell activation, a more robust anti-tumor response in vivo, and the eradication of tumor cells [7].

### 2.1. CART Cell Structure

To date, four generations of CART cells are known. In the first generation of CART cells, the extracellular moiety for antigen recognition is most commonly a single-chain variable fragment of an antibody (hinge or spacer structure). This structure is anchored to a transmembrane component with an attachment, on the intracytoplasmic cell membrane, to T-cell signaling molecules. First-generation CARs only had an intracellular signaling domain of CD3ξ (cluster differentiation 3ξ) to activate T cells [20,21]. The receptors that contained single-chain variable fragments recognized antigens independently of the MHC (major histocompatibility complex) and comprised target cell surface proteins, carbohydrates, and glycolipids [22]. First-generation CARs can induce a cytotoxic antitumor response when T cells are activated. They can eliminate cancer cells effectively through secreting granzyme and expressing FasL (Fas cell surface death receptor ligand) and tumor-necrosis-factor-related apoptosis-inducing ligand (TRAIL) [23]. After repeated exposure to antigens, they do not result in T cell expansion [24]. In the second generation, Maher et al. interposed a costimulatory signaling domain 4-1BB (CD137 or tumor necrosis factor receptor superfamily 9-TNFRSF9) or CD28 costimulatory signaling domain between both the transmembrane and T-cell activating domains of a ζ chain fusion [25]. CD28, in association with co-stimulatory CD3ξ, represents the intracellular signaling domain that allows T cell activation and proliferation [23,26]. Third-generation CARs comprise three intracellular signaling domains: CD3ξ and two costimulatory domains, CD28 and 4-IBB [23]. In fourth-generation CARs, T cells are redirected to improve cytotoxicity and edit the immune system with the inducible release of a transgenic payload (TRUCKs). CAR-redirected T cells deliver a transgenic product (‘payload’) to the targeted tumor tissue, and it is made of a nuclear factor of activated T cell (NFAT) domain, a suicide gene, or signaling domain that induces the production and release of proinflammatory cytokines, like Interleukin-7 (IL-7), IL-18, or IL-12 [27]. IL-12 improves T cell activation, modulates the immunological and vascular TME, and recruits additional immune cells.

To date, tisagenlecleucel and axicabtagene ciloleucel are the only CD19-targeted CART cells approved, and they belong to the second generation [28].

### 2.2. Toxicities

Principal CART cells’ side effects are related to massive T cell activation. Different grading systems are currently used for CART-cell-related toxicity (ASTCT, CARTOX, and CTCAEv5.0), with significant differences between grading systems and the potential to undertreat or overtreat symptoms [29,30]. 

#### 2.2.1. On-Target On-Tumor Toxicity: Cytokine Release Syndrome

According to the American Society for Transplantation and Cellular Therapy (ASTCT), cytokine release syndrome (CRS) is defined as an excessive response following any immune therapy that determines the activation or recruitment of endogenous or infused T cells and/or other immune effector cells [31]. The syndrome is related to rapid immune reactions driven by the massive release of cytokines, such as IFN-γ, IL-6, and IL-10. It develops several days after CART-cell infusion; manifests with fever, hypotension, and tachycardia; and can potentially progress to hemodynamic instability, resulting in end-organ injury [32,33,34].

Effective therapeutic options are tocilizumab, an anti-IL6 receptor antibody, and corticosteroids [35,36,37].

#### 2.2.2. On-Target Off-Tumor Toxicity

On-target off-tumor toxicity represents a major obstacle to successful cancer immunotherapy. It is induced by a direct attack on healthy tissue cells expressing the targeted antigen. Considering the strength of redirected T cells, toxicity in non-pathogenic tissues can be highly damaging. It is typically managed in two ways: restricting antigen choice with co-stimulation-only CARs that lead to cytotoxic activation only when they meet a tumor cell or by accepting non-tumor antigen expression and the potentially associated severe toxicities [38].

#### 2.2.3. Off-Target Toxicity

Off-Target toxicity occurs when the transduced T cell population attacks an antigen other than the targeted one or activates themselves regardless of their specificity. The off-target recognition of cross-reactive antigens has not been detected in CART-cell trials to date [39].

#### 2.2.4. Neurotoxicity: ICANS (Immune-Effector-Cell-Associated Neurotoxicity Syndrome)

ASTCT defined ICANS as a pathologic process involving the central nervous system following any immune therapy that is characterized by the activation or recruitment of endogenous or infused T cells and/or other immune effector cells [31].

The incidence of immune-effector-cell-related neurotoxicity syndrome is closely correlated with CRS; it is more common in patients with B-ALL than in those with other diseases that receive CD19-specific CART-cell therapy [40,41]. The median time to the beginning of neurotoxicity ranges from 4 to 10 days after CART cell infusion with a median duration of 5–14 days. Symptoms can comprise headache, lethargy, impaired attention, dysgraphia, apraxia, aphasia, agitation, tremor, encephalopathy, and seizures. Treatment for neurotoxicity consists of supportive care, with or without corticosteroids and/or cytokine-directed therapies, based on the presence of active CRS [42].

#### 2.2.5. Immunogenicity: Anaphylaxis

The potential immunogenicity of antigens used in engineered T cells may lead to severe anaphylaxis [43,44]. It might be associated with the release of IgE (immunoglobulin E) antibodies specific to the CAR-modified T cell product or to the in vitro T cell production process. High levels of tryptase, pathognomonic mast cell activation, and cytokines, particularly IL-6, IL-10, and IL-2, were observed in a clinical study given repeated infusions of the CART cells [45].

In some cases, CART cell therapy may be characterized by limited efficacy against solid tumors. One of the principal causes of a lacking or weak response is poor T cell expansion and T cell exhaustion triggered by co-inhibitory pathways [46]. Therefore, a combination of CART cells and immune checkpoint inhibitors (ICIs) is thought to lead to strong immune responses.

## 3. Tumor Microenvironment (TME)

Tumors’ behavior is determined by their intrinsic characteristics and by their interaction with components of the TME. The TME is a highly structured ecosystem constituted by an immune and vascular part and extracellular matrix (ECM) fraction. The immune component consists of different immune cells, such as T cells, natural killer (NK) cells, dendritic cells (DCs), myeloid-derived suppressor cells (MDSCs), and tumor-associated macrophages (TAMs). M1 macrophages are pro-inflammatory phenotypes and are able to initiate and maintain inflammatory responses, secreting pro-inflammatory cytokines, activating endothelial cells, and inducing the recruitment of other immune cells into the inflamed tissue. On the other hand, M2 macrophages promote the exhaustion of inflammation, phagocytose apoptotic cells, drive collagen deposition, ensure tissue integrity, and release anti-inflammatory mediators [47]; the vascular component comprises blood and lymphatic endothelial cells that communicate with one another and with the different cancer cells present in the TME.

The ECM fraction consists of complex collagen fibers and other glycoproteins; the ECM is formed by laminin, fibronectin, elastin, and collagen. ECM aggregates in solid tumors are more than 60% of tumor mass [48]. Cancer-associated fibroblasts (CAFs) and mesenchymal stem cells (MSCs) represent the stromal component [49,50,51].

The main suppressive factors of solid-tumor TME factors are MDSCs, Tregs (regulatory T cells), chemokines, and cytokines produced by immune cells that activate transcription factors such as AP-1, NFκB, and STAT3, which support malignant cell proliferation and survival [52,53]. All TME components promote inflammation, induce the formation of new blood vessels, support tumor growth, avoid destruction by the immune system, evade cell death, and induce the activation of invasion and spreading to other parts of the body. Simultaneously, they stimulate the recruitment of immune cells at the site of neoplastic processing.

Principal components of the immune response are represented by several subpopulations of T lymphocytes. Th1 (T helper 1 lymphocytes) cells producing INFy (interferon y) have antitumor activity, stimulate cytotoxic T lymphocytes and NK cells, and recruit macrophages for their anti-tumor activity. On the other hand, Tregs have an inhibitory effect on Th1 lymphocytes, promoting immunosuppression and tumor cell growth. Th2 and Th17 produce cytokines (e.g., IL-4, IL-5, IL-13, IL-17, IL-21, and IL-22) that contribute to enhanced tissue inflammation and boost tumor growth. Regulatory T lymphocytes (CD4, CD25) inhibit the activity of other lymphocytes, mainly T cytotoxic ones [54,55,56].

The amount of T cells in neoplastic tissue seems to have a prognostic value [57]. In particular, one of the prognostic factors is represented by the presence of CD8-positive cytotoxic T lymphocytes in the TME before starting treatment because it plays one of the most important roles in the immune response [58,59].

Based on the cellular composition of lymphocyte infiltration cells in the TME, solid tumors distinguish three immune profiles. The “hot” tumors, or inflammatory tumors, are strongly infiltrated by a wide variety of cells: B lymphocytes, CD4-positive and CD8-positive T cells, Treg lymphocytes, macrophages, fibroblasts, and MDSCs, with many inflammatory signals and strong leukocyte infiltration. Additionally, these tumors are characterized by the presence of intra-tumor chemokines (e.g., CXCL9, CXCL10, CCL5) [60]. They have high cell density inside the tumor regardless of cell density in the tumor periphery. The “cold” tumors, or non-inflammatory tumors, are characterized by low cell density inside and outside the tumor and inflammatory signs. The “cold tumors” can be distinguished into two types: so-called “immune desert” tumors, with the presence of macrophages, Treg lymphocytes, and MDSCs in the absence of infiltration or activation of T lymphocytes, and tumors with “immune exclusion” characterized by high immune-cell density at the periphery or inside the stromal tissue and low density in the core of the tumor [61]. The immune cells do not infiltrate the parenchyma of the tumors due to the presence of the barrier character of fibroblast cells surrounding the tumor parenchyma [62].

The main differences between hot and cold tumors are summarized in Table 1.

Many solid tumors show hyperoxygenation, resulting in areas of permanent or transient hypoxia [63,64]. Hypoxia promotes the expression of HIF-1 (Hypoxia-Inducible Factor 1), for which overexpression highly correlates with worse prognosis for several solid cancers. Hypoxia stimulates the growth of tumor cells, inducing changes in the TME. By cross-talking with the nuclear factor-kB pathway, HIF-1a (Hypoxia Inducible Factor 1 Subunit Alpha) is able to modulate principal inflammatory functions in myeloid cells; promote the proliferation of regulatory lymphocytes that inhibit the differentiation of effector T cells [65,66], cell-cycle changes, and heat stress response; and play a critical role in high radioresistance [64]. 

Low levels of oxygen lead cells to reprogram energy metabolism. Cancer cells exhibit repressed mitochondrial respiration and a high rate of glucose uptake even in the presence of oxygen. This metabolic process, known as both the Warburg effect and aerobic glycolysis, promotes survival and long-term maintenance for tumor cells [67,68,69]. 

In dendritic cells and macrophages, pro-inflammatory stimuli induce the shift to aerobic glycolysis. In macrophages, the glycolytic phenotype promotes polarization from anti-inflammatory/pro-resolving M2 to the classically activated pro-inflammatory M1 and the production of many inflammatory mediators.

HIF-1α activates the secretion of pro-angiogenic factors such as VEGF-A. VEGF-A inhibits the release of inter-endothelial and vascular cell adhesion molecules (ICAM-1, VCAM-1) and hinders the migration of CART cells across the vascular endothelial cell barrier into the tumor tissue [70,71].

Hypoxia is associated with a poor prognosis and resistance to chemotherapy and radiotherapy [72]. 

The passage and activation of CART cells depend on tumor blood vessels; therefore, when CART cells are far away from blood vessels, they cannot reach tumor tissues, resulting in therapy resistance. At the same time, in a hypoxic TME, tumor cells produce cytokines, which are recruited to immunosuppressive cells to deplete CART cells, preventing their differentiation into effector memory cells [73].

To overcome hypoxia-related resistance, recently, some authors developed a dual oxygen-sensing approach to detect cell targets in hypoxia conditions. CAR cells were fused with an oxygen-dependent degradation domain (ODD) of HIF1α (CAR-ODD) [74,75]. Afterward, the CAR was implemented, adding the CAR’s promoter in the long terminal repeat (LTR) enhancer region of hypoxia-responsive elements (HREs), leading to HIF1α-mediated transcription of the CAR (HypoxiCAR). In this way, HypoxiCAR is able to detect surface antigens only in hypoxia conditions (0.1% O_2_) [76].

There are several trials that investigate the use of CART cells therapy in solid tumors such as glioblastoma, prostate cancer, and pleural cancer [77,78,79,80,81,82].

## 4. Radiotherapy

Radiotherapy (RT) has an anti-tumor effect both, modifying the TME and inducing DNA damage [83,84,85]; the mutations induced by RT enhance the expression of TAA, activate and increase endogenous-target antigen-specific immune responses, and favor the survival of antigen-positive cells [86], promoting CART cell engagement and expansion and modifying the inflammatory TME with an antigen-dependent mechanism [87,88]. 

Therefore, the induction of target antigen expression favors the efficacy of CART cell therapy after RT [89,90]. The immune cells present different sensitivity to irreversible damage induced by RT.

RT can overcome barriers induced by immunosuppressive TME and radioresistant tumor cells [91]. 

In fact, RT activates immune cells, increases the density of TILs, and facilitates recognition of tumor cells by T cells with induced immune-mediated cell death through the regulation of multiple cytokine signaling, including TNF, IL-1b, IL-10, and transforming growth factor beta (TGFβ) [92,93]. 

Ionizing radiation leads to the impairment of vascular cells, but those remaining or surrounding the irradiated tissues produce proinflammatory change [94]. On blood vessels, RT induces an increase in the production of the integrins ICAM-1 and VCAM-1 in vascular endothelial cells and promotes the transition of CART cells across the vascular endothelium into the tumor tissue [95,96]. 

On the other hand, RT supports immunosuppression in the tumor immune microenvironment with increased Tregs infiltration in the irradiated tumor zone. Tregs, with high Akt (Akt serine/threonine-protein kinases) expression, are more radioresistant than other T cell subsets and hinder effector T cells and CART cells’ activity [97,98,99].

Different fractionation radiation doses, rather than the total dose, can determine different effects on the TME [100,101].

Several studies showed that the best fractionation schedule to use for immunotherapy combinations was 8 Gy × 3 Fx [102,103,104,105].

In a randomized phase 2 clinical trial, Schoenfeld et al. and Monjazeb et al. evaluated metastatic NSCLC and colorectal cancer, respectively, two different RT schedules known to increase immune responses: patients received ICIs alone or with repeated low-dose RT (0.5 Gy twice per day) or hypo-fractionated radiotherapy 24 Gy (8 Gy × 3 Fx). In both cases, a reduction in CD8+ T cell populations into the TME following directed radiotherapy was observed, especially with higher radiation doses [104,105].

Meng et al. investigated that a large single dose (20 Gy) or at 2 Gy in 10 fractions (10 × 2 Gy) modifies the immunosuppressive TAMs phenotype, but some studies showed an opposite effect of short-course or low-dose RT on TAMs [106].

Pancreatic cancer is considered a cold tumor, with a low mutation burden and a low response to T cell infiltration, which indicates its weak response to immunotherapy. In a preclinical study, sialic acid Lewis-a (sLeA) CART-cell (sLeA)-expressing pancreatic cancer tumor-bearing mice were associated with 2 Gy single low-dose radiotherapy. In this trial, the authors reported that local radiation led to a higher sensitivity of pancreatic cancer cells to activated CAR-T cells, and that combination therapy increased the sensitivity of sLeA– and sLeA+ tumor cells [102]. 

Even Tregs cells’ infiltration grade based on the dose delivered was observed [104,105,106,107,108,109].

High-dose radiation therapy (HDRT, >10–20 Gy/fractions) converts immunologically “cold” tumors into “hot” by determining the death of cancer cells, resulting in the release of proinflammatory mediators, such as damage-associated molecular patterns (DAMPs), high-mobility group box 1 protein (HMGB1), ATP, and calreticulin, which stimulate Toll-like receptor 4 (TLR4). 

The combination of HDRT and ICIs hinders Tregs and mitigates exhaustion. In general, higher RT doses induce necrosis and massive inflammation [105,106], increase the proportion of M1-like macrophages in the TME, and reduce the proportion of immunosuppressive M2-like macrophages and MDSCs post conventionally fractionated RT (2 Gy/fraction). MDSCs have an immunosuppressive effect and can prevent anti-tumor immune responses [110,111,112].

On the contrary, low-dose RT (LDRT, >5–10 Gy/fractions) can reprogram the TME, encouraging the infiltration of effector immune cells and modulating the stroma to allow tumor eradication.

Klug et al. were the first to demonstrate, in a mouse model of neuroendocrine pancreatic cancer, that radiation doses ranging from 0.5 to 2 Gy induced de novo T cell homing into immune “cold” tumors, promoting M1 macrophage transition, normalization of tumor blood vessels, and increasing infiltration of adoptively transferred T cells [103,113]. LDRT so polarizes pro-tumor M2 macrophages to the antitumor M1 phenotype, increases the infiltration of CD4+ T cells and NK cells, and downregulates TGF-β inhibitory cytokines [114]. 

Delivering a 1 Gy × 2 dose alone in a lung adenocarcinoma model showed control of tumor growth and extended survival. Moreover, the efficacy of ICIs was significantly improved when associated with LDRT. At the same time, LDRT decreased cancer-associated fibroblasts (CAFs) in the tumor stroma context and allowed for increased infiltration of immune cells [115].

LDRT (0.1 Gy) was found to block the transformation by inducing antioxidants and reducing ROS (reactive oxygen species) [116].

LDRT is usually used for hot tumors, HDRT for cold ones.

Even when CART cell therapy is used in association with RT, the fractionation dose should be defined [116,117].

Several studies suggested that LDRT makes the tumor more sensitive to CART and may be a better support to CART cell treatment based on the radioimmunological effects. LDRT increases the expansion of infused T cells [118,119]. 

Rotte et al. proved that there exists a threshold dose beyond which, despite enhancing the dose, further dose escalation is unlikely to result in better objective response rates (ORRs) of CARTs [120]. 

Moreover, LDRT may play an immunomodulatory effect when combined with ICIs. This effect was defined as “radscopal effects” by James Welsh and refers to the systemic antitumor effects based on the combination of HDRT to the primary tumor and LDRT to the metastatic site in patients undergoing immunotherapy [113]. 

### 4.1. LATTICE Therapy

An emerging and innovative approach, defined as the “LATTICE technique”, represents a particular form of spatially fractionated radiation therapy that allows for the delivery of heterogeneous radiation doses. 

In clinical studies, 30 patients were treated with the palliative “metabolism-guided” lattice technique, a therapeutic approach in which high doses are delivered to vertices (“peaks”) allocated according to 18F-FDG uptake and maintaining low doses to the tumor periphery (“valleys”). This permits the mobilization of the systemic immune-mediated tumor response [121]. The LATTICE technique will be explained in the proper section (Section 4.1.1).

The effect of LATTICE radiotherapy may be increased by association with ICIs. The high doses delivered to vertices lead to the release of DAMPs with immunogenic cell death. The valleys’ low doses maintain the residual blood flow and permit DAMP circulation, immune cell homing, and activation. Jiang et al. reported that only the association of RT and ICIs increases tumor immunogenicity and immune response both in irradiated and non-irradiated areas (abscopal effect) [122,123]. 

The effects of different radiation doses and ICIs on tumor cells are summarized in Figure 1.

Because CART cell therapy is a form of immunotherapy, according to this knowledge, a heterogeneous radiation dose may be associated with CART therapy and ICIs to overcome a resistant TME.

#### 4.1.1. LATTICE Technique

LATTICE therapy, a variant of spatially fractionated radiation therapy (SFRT), is a radiation therapy technique that allows one to deliver high lethal radiation doses to a large portion of tumor volume with OAR sparing.

The two main form of SFRT with photons are GRID radiotherapy (GRID RT, a 2-dimensional technique) and lattice radiotherapy (LRT)—a 3D configuration of GRID that allows one to deliver a high dose peak with a spherical shape within the tumor, surrounded by a lower-dose valley.

The GRID technique attempts to achieve a differential dose distribution using cerrobend blocks with holes inside, positioned into the gantry head. By alternating blocks and holes, this radiation field array generates a two-dimensional dose distribution, characterized by foci of high radiation doses (peaks) separated by low-dose areas (valleys).

With the advent of medical linear accelerators producing megavoltage (MV) photon beams and multileaf collimators (MLCs), the original GRID technique was dropped out and was replaced by the LATTICE radiotherapy technique, in which virtual grid blocks and rigid spatial dose fractionation was reproduced using treatment planning systems (TPSs) with the placement of high dose spheres, called vertices, within the tumor [124]. Usually, each vertex is about 1 to 2 cm in diameter with a distance of 2–3 cm from each other (center-to-center distance). Generally, a dose >15 Gy in a single fraction or a biological equivalent if fractionated is prescribed, since lower doses could be ineffective to achieve a satisfactory decrease in tumor volume [121]. 

The optimal LATTICE design is not well-defined because there is not any theory supported by evidence on this issue. In a recent study, Vertex placement did not follow a geometric pattern according to classic SFRT [125]. Because regions with different SUVs within a bulky mass could show different cell growth rates, oxygenation, and tumor microenvironments, vertices were placed in different 18F-FDG-uptake areas within the bulky disease. A “Photopenic PET Area” (PPA) showing a necrotic core and a “Super-Avid” PET Area (SAPA) with SUV > 75% SUVmax were detected. Vertices were positioned between SAPA and the remaining part of the APA (“Avid” PET Area (APA) with SUV > 2.5) [123].

Non coplanar beams or VMAT (Volumetric Modulated Arc Therapy) can be used to perform a dosimetric evaluation [126,127].

## 5. Conclusions

CART cells have revolutionized the treatment of several hematological malignancies, but there is little evidence on solid tumors. Based on this narrative review, promising results are also being obtained from CART cell therapy in solid tumors when combined with radiotherapy. Whereas heterogeneous radiation doses can edit the TME and enhance TAA expression, ICI therapy and CART cell therapy can determine a stronger T cell activation and a more robust anti-tumor response. Taken together, this knowledge provide the basis for a new emerging approach therapy that combines SFRT, ICIs, and CART cells.

## Figures and Tables

**Figure 1 jpm-13-01261-f001:**
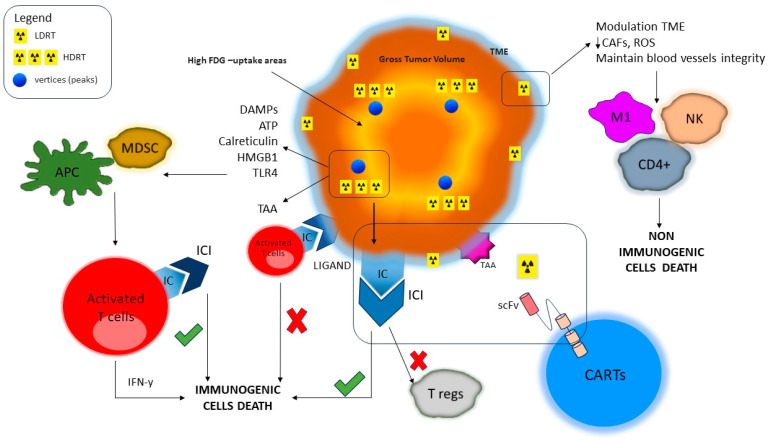
Effect of differentiated doses on tumor cells. High doses of radiation therapy lead to immunogenic cell death through both DAMPs, ATP, calreticulin, HMGB1, and TLR4 production, and the associated tumor antigens. This process results in the maturation of dendritic cells and the activation of T cells. Low doses of radiation therapy lead to non-immunogenic cell death through macrophage polarization (M2 → M1) and recruitment of CD4+ cells and NK cells. LDRT reduces ROS and maintains blood vessel integrity, allowing the movement of CART cells and cell effectors across the vascular endothelium into the tumor tissue. Tumor cells and activated T cells express on a surface immune checkpoint (IC). The link between the IC and the corresponding ligand inhibits T cells from killing tumor cells. The use of immune checkpoint inhibitors (ICIs) allows TC cells to kill tumor cells. The link IC–ICI blocks T reg activity. CARTs can bind antigen-associated tumors (TAAs) and induce tumor cell death. T cell exhaustion is one of the limitations of CART effects. The use of ICIs can elicit a durable response; LDRT increases the expansion of infused CART. HDRT increases immunosuppressive cells.

**Table 1 jpm-13-01261-t001:** Characteristics of hot and cold tumors.

	T Cell Infiltration	Mutation Burden and PD-L1 Expression	Response to Immunotherapy	Example
Hot Tumor	- High cell density inside the tumor - Many inflammatory signals	High mutation burden and high expression of PD-L1	Sensitive	Melanoma, non-small-cell lung cancer, and cancers of the bladder, head and neck, kidney, liver
Cold Tumor	- Low cell density inside and outside the tumor - Low inflammatory signs	Low mutation burden and low expression of PD-L1	Resistant	Breast, ovary, prostate, pancreas, brain glioblastoma

## Data Availability

Not applicable.

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
