# Peer review of "Modulation of Radiation Doses and Chimeric Antigen Receptor T Cells: A Promising New Weapon in Solid Tumors—A Narrative Review"

_jpm, 2023, doi:10.3390/jpm13081261_

Round 1

Reviewer 1 Report

This review is very difficult to read. Most of it is written as one sentence summaries of different papers just thrown onto a page without any attempt to make them into coherent paragraphs. Some parts are even left in list format, like lines 155-161. The abstract said the purpose of this review was to clarify the knowledge to date on CART cell therapy and the combination with RT in solid tumors, but nothing was really made clear in this paper. I'm not a physician or a biologist, and there were many biological terms that were used without any introduction or explanation, so a non-expert can't learn anything from this review. The section on Radiotherapy is particularly poor; many of these studies were specific to a particular cancer but that's never mentioned for any of them. I wouldn't consider this review worth publishing unless there was a major revision to make it more readable. 

There are many English language errors like wrong verb choice which should be pointed out by any English language word processing program like Microsoft Word or typing assistant like Grammarly.

Author Response

This review is very difficult to read. Most of it is written as one sentence summaries of different papers just thrown onto a page without any attempt to make them into coherent paragraphs. Some parts are even left in list format, like lines 155-161. The abstract said the purpose of this review was to clarify the knowledge to date on CART cell therapy and the combination with RT in solid tumors, but nothing was really made clear in this paper. I'm not a physician or a biologist, and there were many biological terms that were used without any introduction or explanation, so a non-expert can't learn anything from this review.The section on Radiotherapy is particularly poor; many of these studies were specific to a particular cancer but that's never mentioned for any of them. I wouldn't consider this review worth publishing unless there was a major revision to make it more readable. 

R: The list format was removed. It is a topic still little explored.

The manuscript was modified to make it more understandable. See the lines 58-62, 136-137, 232,338-343.

The radiotherapy section was improved with Lattice technique paragraph, as request by Revisor 2. See the line 311-336

Reviewer 2 Report

This is a well-researched and well-written review on one aspect of LATTICE therapy. The article could be a good resource for investigators who want to know more about the effectiveness of the LATTICE therapy from an immunology point of view.

General comments:

Adding a subsection focusing on the LATTICE therapy and providing more details on this technique will make the review more focused and valuable for potential readers because the abstract claims that the current study is for this technique.

Minor comments:

[Line 68] Spell out “MHC”

[Line 169] “Th1 producing INFy, have antitumor activity” should be “Th1 producing INFy has …”

[Line 170] In “On the other hand, have an inhibitory effect,” what “have an inhibitory effect”?

Overall, it is ok. See a few minor editorial corrections given in the comments to the authors section.

Round 2

Reviewer 1 Report

The authors made some modifications to the text but these edits did not address my primary concerns so I have listed those in detail here:

-          Abstract: what is LDRT? Why do you need to introduce an acronym for “high dose radiation therapy”? How much dose is considered HDRT, instead of just RT? The term is used again in line 251 but the text still does not define how much dose is required for HDRT.

-          Line 36: “become activate” doesn’t make sense

-          Liine 37: “foreign invade” doesn’t make sense

-          Line 38: “prevent treatment resistant minimizing tumor burden” doesn’t make sense

-          Line 44: “homing to the tumor” doesn’t make sense

-          Line 60: define MHC here not in line 69

-          Lines 64-83: lots of undefined biology terms here, like CD3E, FasL, 4-1BB (or is it 4-IBB as in line 78?), CD137, CD28. Also switches back and forth between CART, CAR T, and CAR/CARs; should pick one and be consistent through the entire manuscript. Are all four generations of CART cells currently in use? Or only 4th generation?

-          Line 91: what is “it” here? Is “it” Cytokine release syndrome? Should state that.

-          Line 100: what is “it” here? This time I don’t even have a guess.

-          Line 109-110: this is a sentence fragment, not a full sentence.

-          Line 117: what is CRS?

-          Line 118: what is B-ALL?

-          Line 127: what is IgE?

-          Line 136: this sentence seems out of place, why is it in the Toxicity section? Maybe it should go in the paragraph starting with line 186 which discusses solid tumors?

-          Line 142-144: this is a sentence fragment, not a full sentence.

-          Line 153: what are Tregs?

-          Line 160: Th1, INFy are undefined

-          Line 166-168: these sentences are on RT, why are they in the TME section?

-          Line 167: what is Akt?

-          Line 180: “distinguished two types” doesn’t make sense

-          Line 187: what is HIF-1?

-          Line 220: “favorite CART cell” doesn’t make sense

-          Lines 231, 232, 235: what disease types or sites were these studies performed on?

-          Line 241: “favourites the transition” doesn’t make sense

-          Line 261: What qualifies as LDRT? Is it the total dose or the dose per fraction that is low? Or both?

-          After reading lines 255-282, I don’t know for which tumors HDRT is better and for which LDRT is better, or what qualifies as HDRT/LDRT.

-          Why are lines 283-293 not part of the new LATTICE section that starts on line 311?

-          Line 315: “to deliver spherical” doesn’t make sense

-          Line 317: “positionated” is not a word. And it doesn’t alternate blocks and holes, it has holes formed in the block.

There are many English language errors (some detailed above) like wrong verb choice which should be pointed out by any English language word processing program like Microsoft Word or typing assistant like Grammarly.
